# Application of the JISP16 potential to the nucleon induced deuteron breakup process at E=13 MeV and E=65 MeV

Volodymyr Soloviov[1*], Jacek Golak[1], Roman Skibiński[1],
Kacper Topolnicki[1] and Henryk Witała[1]

1 M.Smoluchowski Institute of Physics, Jagiellonian University, PL-30348, Kraków, Poland

★ volodymyr.soloviov@doctoral.uj.edu.pl

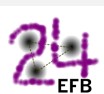
*Proceedings for the 24th edition of European Few Body Conference,
Surrey, UK, 2-6 September 2019*

## Abstract

**The JISP16 nucleon-nucleon potential has been applied to investigations of the nucleon induced deuteron breakup reaction at the incoming nucleon laboratory energies E = 13 MeV and E = 65 MeV. We have found that for the studied process the JISP16 force gives a description of the exclusive cross section, which is generally similar to the ones obtained with the standard realistic nucleon-nucleon AV18 interaction. However, there are some regions of the phase space where the differential cross sections predicted by the JISP16 and AV18 models, differ by more than 100 %. These special kinematical configurations may possibly be useful to refit the JISP16 force parameters.**

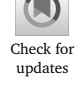

## 1 Introduction

The JISP16 potential [1] is a model of the nucleon-nucleon (NN) interaction derived within the inverse scattering method. Its free parameters have been fixed by fitting to the NN phase shifts, as well as to the energies of bound and excited states of nuclei up to $^{16}$O. This method of fixing parameters was introduced to study many-nucleon systems without using many-nucleon interactions. The JISP16 model works well in the nuclear structure calculations [2], but it was shown in [3] that it leads to disagreements with data and theoretical predictions based on standard nuclear forces when applied to the photoabsorbtion processes on light nuclei. Also recently Ref. [4] has shown that the JISP16 potential fails in the description of some observables in the elastic nucleon-deuteron scattering process. The reason for such behaviour was traced back to the P-wave components of the JISP16 model. In this contribution we investigate usefulness of the nucleon-deuteron breakup reaction in improving the JISP16 potential. The results for the AV18 force are used as reference values. The expected three-nucleon force effects at the incoming nucleon energy $E$=13 MeV are negligible and at $E$=65 MeV remains

below approx. 5% at the most of the phase space, reaching approx. 20% only for some specific configurations [5].

The final kinematical configuration for the studied $n + d \rightarrow n + n + p$ reaction is specified by five variables. We have chosen the following variables: the polar and azimuthal angles of the momentum vector of nucleon 1 ($\theta_1, \varphi_1$) and the momentum vector of nucleon 2 ($\theta_2, \varphi_2$) as well as the energy of nucleon 1 ($E_1$). In some cases this set gives two physical solutions for the energy of nucleon 2 ($E_2$), so the arc length $S$ for the kinematical locus in the $E_1 - E_2$ plane is used instead of $E_1$ to assure a unique definition of the three-body kinematics. We choose nucleons 1 and 2 to be neutrons.

The fact that the phase space is spanned by the ($\theta_1$, $\varphi_1$, $\theta_2$, $\varphi_2$, $S$) set of variables allows us to study various dynamical and kinematical aspects of the three-nucleon (3N) observables [5, 6]. This, in turn, gives a possibility of a more systematic and detailed analysis of the nucleon-nucleon interaction than is available e.g. in elastic nucleon-deuteron scattering [7].

## 2 Formalism and methods

The Faddeev equation has been used to compute the transition amplitude and, finally, the differential cross section for the nucleon-deuteron breakup process. Neglecting the 3N interaction the Faddeev equation for an auxiliary state $T|\phi\rangle$ reads [8]:

$$T|\phi\rangle = tP|\phi\rangle + tPG_0T|\phi\rangle \,, \tag{1}$$

where the initial state $|\phi\rangle$ consists of a deuteron and a relative momentum eigenstate of the projectile nucleon, $P$ is a permutation operator, $G_0$ is the free 3N propagator and $t$ is the solution of the Lippmann-Schwinger equation for the two-nucleon t-matrix which depends on the NN interaction model used.

We chose the incoming nucleon energies to be $E = 13$ MeV and 65 MeV. Firstly, for each energy, we solve Eq.(1) twice, using the JISP16 and the AV18 potentials [9]. Next we scan the whole phase space with a grid of 45 points for the $\theta_1$ and $\theta_2$ polar angles in the range (0°, 180°), 45 points for the relative azimuthal angle $\varphi_{12} = |\varphi_2 - \varphi_1|$ in the range (0°, 180°) a fine step of 0.1 MeV along the arc length $S$ for the unpolarized cross section calculations.

This scanning with a subsequent extraction of exclusive observables is done separately for the JISP16 and the AV18 NN potentials.

In order to quantify the discrepancy between the predictions based on the two different models of the NN interaction, we calculate $\Delta SIGS$ - a maximum of relative difference for the cross sections over $\varphi_{12}$ and $S$ for fixed ($\theta_1, \theta_2$):

$$\Delta SIGS \equiv \Delta SIGS(\theta_1, \theta_2) = \max \left[ \frac{Obs_{JISP16} - Obs_{AV18}}{\frac{1}{2}(Obs_{JISP16} + Obs_{AV18})} \right]_{\{\varphi_{12}, S\}} \,, \tag{2}$$

where $Obs = \frac{d^5\sigma}{d\Omega_1 d\Omega_2 dS}(\theta_1, \phi_1, \theta_2, \phi_2, S)$.

In our calculations we apply also additional threshold cuts to avoid kinematical configurations with very low $E_1$ or $E_2$ nucleon energies or with the cross section values unmeasurable in practice.

## 3 Results

In order to present results of scanning the five-dimensional space in a convenient way we prepared a set of three-dimensional maps. Maps show a dependence $\Delta$SIGS, $\varphi_{12}$ and $E_1$ on the polar scattering angles $\theta_1$ and $\theta_2$.

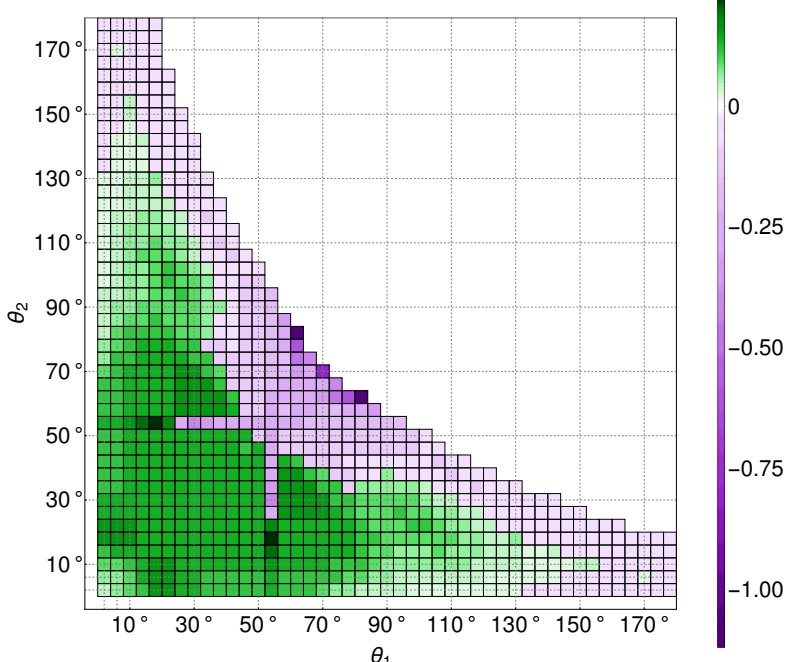

Figure 1: The distribution of $\Delta$SIGS for the deuteron breakup at $E=$ 13 MeV.

The map given in Fig. 1 shows a dependence of a maximal relative difference of the cross sections $\Delta$SIGS on the scattering angles $\theta_1$ and $\theta_2$. It allows us to identify phase space regions with a strong deviation between cross sections based on the JISP16 and the AV18 models. In order to determine the remaining kinematical variables for configurations where the maximal $\Delta$SIGS over $\varphi_{12}$ and $S$ is observed, we prepared additional maps (Figs. 2 and 3) for the dependence of the relative azimuthal angles $\varphi_{12}$ and energy $E_1$ of outgoing nucleon on the polar scattering angles $\theta_1$ and $\theta_2$. We present here only a map for the energy of outgoing nucleon 1, because an energy for the nucleon 2 is asymmetrical with respect to the diagonal. It can be seen, that at initial energy $E=$13 MeV, in some regions of the phase space the exclusive differential cross section predicted by the AV18 model is bigger than the one based on the JISP16 potential by more than 100%.

The map for the $\Delta$SIGS at 65 MeV in Fig. 4 shows, compared to Fig. 1, fewer discriminative regions allowed by 3N kinematics and the additional energy cuts, but there are also areas where the difference between the cross section values calculated with the JISP16 and the AV18 potentials exceed 50%. It can bee seen in Fig. 5, that these interesting configurations are mostly related to small relative azimuthal angles. The map in Fig. 6 delivers information about corresponding energy of the outgoing nucleon and includes many measurable cases among the interesting configurations.

## 4 Conclusion

In this study, we have found kinematical configurations for the nucleon-deuteron breakup reaction for which the difference between predictions for the exclusive cross section obtained with the JISP16 and with the AV18 force amounts up to 100% (50%) at incoming nucleon energy 13 MeV (65 MeV) . The kinematical variables defining these interesting configurations can be identified with the presented maps. Many such configurations seem to be experimentally accessible.

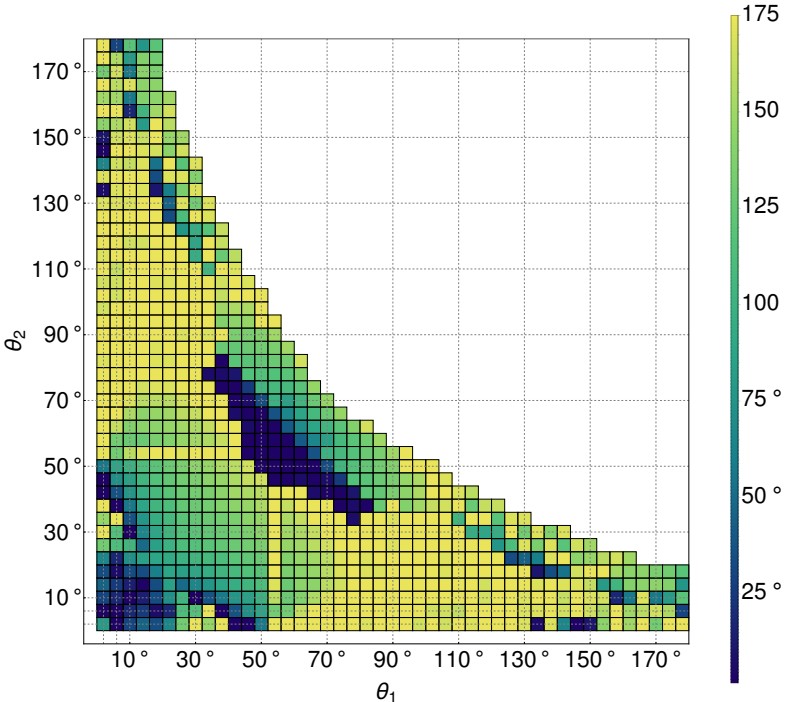

Figure 2: The distribution of the relative azimuthal angle $\varphi_{12}$ for configurations corresponding to the maximal $\Delta$SIGS in the cross section distribution (see Fig. 1) for the deuteron breakup process at $E= 13$ MeV.

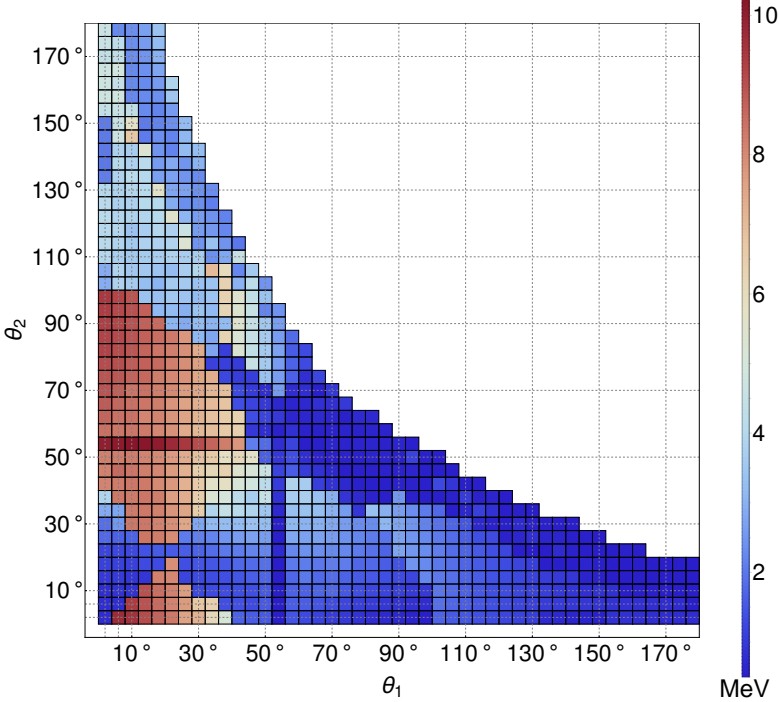

Figure 3: The distribution of the outgoing neutron energy $E_1$ for configurations corresponding to the maximal $\Delta$SIGS in the cross section distribution (see Fig. 1) for the deuteron breakup process at $E= 13$ MeV.

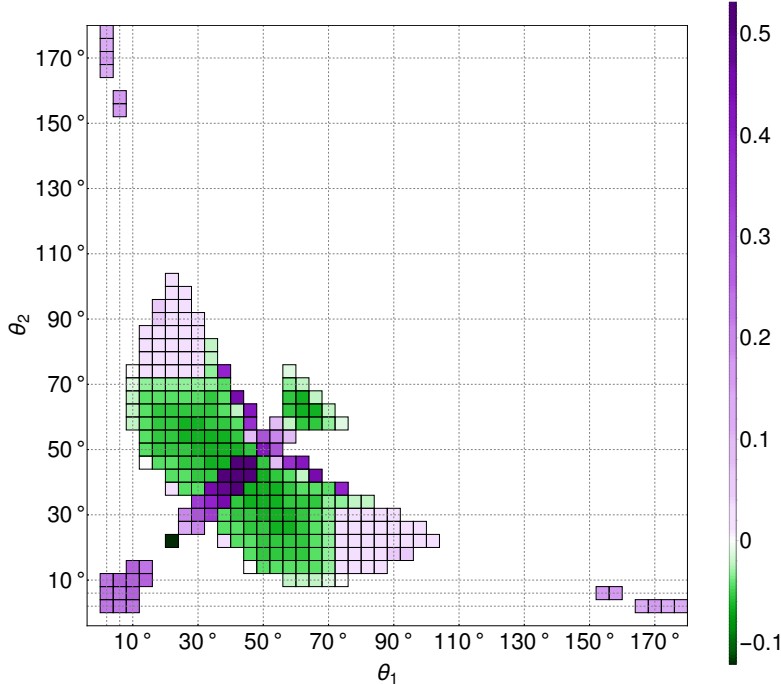

Figure 4: The distribution of ΔSIGS for the deuteron breakup at $E=$ 65 MeV.

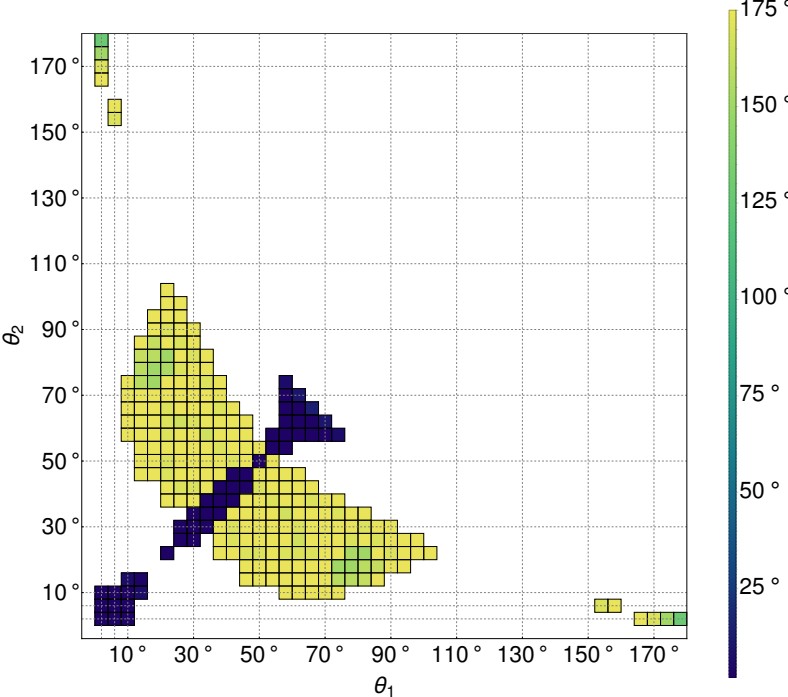

Figure 5: The distribution of the relative azimuthal angle $\varphi_{12}$ for configurations corresponding to the maximal ΔSIGS in the cross section distribution (see Fig. 4) for the deuteron breakup process at $E=$ 65 MeV.

## Acknowledgements

This work is a part of the LENPIC project. The numerical calculations were partially performed on the supercomputer cluster of the JSC, Jülich, Germany.

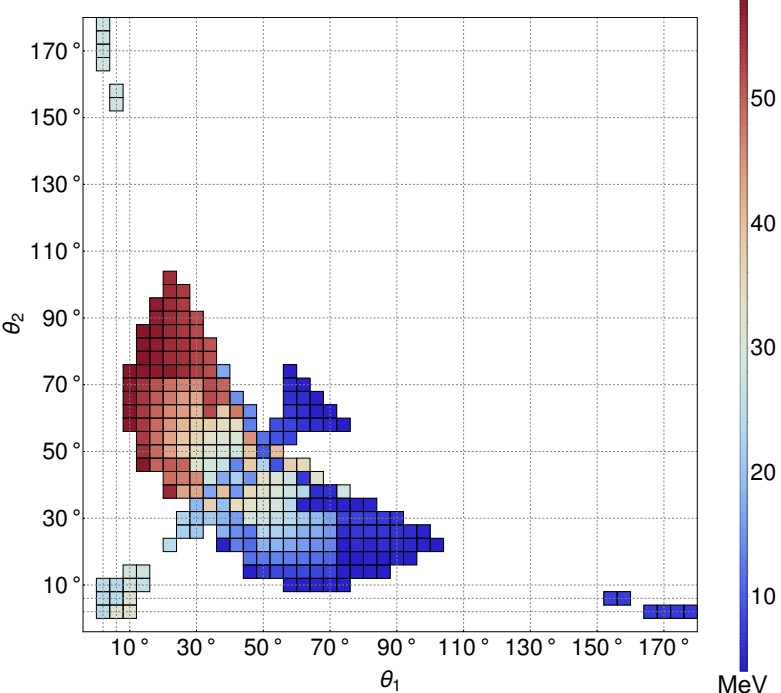

Figure 6: The distribution of the outgoing neutron energy $E_1$ for configurations corresponded to the maximal $\Delta$SIGS in the cross section distribution (see Fig. 4) for the deuteron breakup process at $E$= 65 MeV.

**Funding information** This work is supported by the Polish National Science Centre under Grants No. 2016/22/M/ST2/00173 and 2016/21/D/ST2/01120.

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
