# Peer review of "Application of the JISP16 potential to the nucleon induced deuteron breakup process at E=13 MeV and E=65 MeV"

_SciPost Physics Proceedings, doi:SciPost Phys. Proc. 3, 032 (2020)_

## Round 1 · Referee Report · Winfried Leidemann (Referee 1) · 2019-11-20

Report
The authors make a theoretical study of the neutron induced deuteron breakup.
Two nuclear potential models are employed: AV18 and JISP16 models. As explained by the
authors both potentials are rather different, since, for example, only in case of the AV18
interaction many-nucleon forces are required. Therefore the study undertaken by the authors
is certainly interesting.
The comparison between the two potential models is made for two different neutron energies
of the reaction n + d --> n + n + p. The corresponding exclusive cross section is governed by 5 kinematical variables. In their study the authors find some kinematical settings, where the
difference between both potential models amounts to more than 100%. Such a results is
without question worthwhile to be published.
The authors could consider the following modifications of the manuscript:
(i) The calculation with the AV18 model is made without an additional three-body force.
A comment concerning the effect of the inclusion of such a force would be helpful.
(ii) Figures 2, 3, 5 and 6 do not give units for the colour coding given at the right-hand
side of the figure (for Figs. 2 and 5 the unit degree and for Figs. 3 and 6 the unit MeV is
missing).
(iii) In the introductory part the authors mention a similar previous study for the elastic
neutron-deuteron scattering. They might want to include a further study of the same kind,
where the total photoabsorption cross sections of the 3He and 4He nuclei have been
investigated with quite some discrepancies between both potential models (N. Barnea,
W. Leidemann, G. Orlandini, Phys. Rev. C74, 034003 (2006)).
Two nuclear potential models are employed: AV18 and JISP16 models. As explained by the
authors both potentials are rather different, since, for example, only in case of the AV18
interaction many-nucleon forces are required. Therefore the study undertaken by the authors
is certainly interesting.
The comparison between the two potential models is made for two different neutron energies
of the reaction n + d --> n + n + p. The corresponding exclusive cross section is governed by 5 kinematical variables. In their study the authors find some kinematical settings, where the
difference between both potential models amounts to more than 100%. Such a results is
without question worthwhile to be published.
The authors could consider the following modifications of the manuscript:
(i) The calculation with the AV18 model is made without an additional three-body force.
A comment concerning the effect of the inclusion of such a force would be helpful.
(ii) Figures 2, 3, 5 and 6 do not give units for the colour coding given at the right-hand
side of the figure (for Figs. 2 and 5 the unit degree and for Figs. 3 and 6 the unit MeV is
missing).
(iii) In the introductory part the authors mention a similar previous study for the elastic
neutron-deuteron scattering. They might want to include a further study of the same kind,
where the total photoabsorption cross sections of the 3He and 4He nuclei have been
investigated with quite some discrepancies between both potential models (N. Barnea,
W. Leidemann, G. Orlandini, Phys. Rev. C74, 034003 (2006)).

---

## Round 3 · List of Changes

1. We added (page 1, 1st paragraph of Introduction) an information
together with suitable reference
(new reference [3]) on a previous work by N.Barnea, W.Leidemann, and
G.Orlandini
on application of the JISP force to the photonuclear reactions.

2. At the end of the same paragraph we added a comment on expected
effects of three-body interactions,
which are smaller that the discrepancy observed in the presented studies.

3. In addition, we added missing units (deg and MeV) in figures 2, 3, 5, and 6.

---

## Editorial Decision

published